# Intracellular nucleosomes constrain a DNA linking number difference of −1.26 that reconciles the *Lk* paradox

Joana Segura [1], Ricky S. Joshi[1], Ofelia Díaz-Ingelmo [1], Antonio Valdés [1], Silvia Dyson [1], Belén Martínez-García[1] & Joaquim Roca [1]

The interplay between chromatin structure and DNA topology is a fundamental, yet elusive, regulator of genome activities. A paradigmatic case is the "linking number paradox" of nucleosomal DNA, which refers to the incongruence between the near two left-handed superhelical turns of DNA around the histone octamer and the DNA linking number difference ($\Delta Lk$) stabilized by individual nucleosomes, which has been experimentally estimated to be about −1.0. Here, we analyze the DNA topology of a library of mononucleosomes inserted into small circular minichromosomes to determine the average $\Delta Lk$ restrained by individual nucleosomes in vivo. Our results indicate that most nucleosomes stabilize about −1.26 units of $\Delta Lk$. This value balances the twist ($\Delta Tw \approx +0.2$) and writhe ($\Delta Wr \approx -1.5$) deformations of nucleosomal DNA in terms of the equation $\Delta Lk = \Delta Tw + \Delta Wr$. Our finding reconciles the existing discrepancy between theoretical and observed measurement of the $\Delta Lk$ constrained by nucleosomes.

[1] Molecular Biology Institute of Barcelona (IBMB), Spanish National Research Council (CSIC), Barcelona 08028, Spain. Correspondence and requests for materials should be addressed to J.R. (email: joaquim.roca@ibmb.csic.es)

Cellular DNA is packaged into chromatin via a hierarchical series of folding steps. The basic packaging unit, the nucleosome, contains about 147 base pairs (bp) of core DNA, making near two left-handed superhelical turns around a histone octamer[1]. However, nucleosomes are not uniform and static entities. They can present positional instability, conformational fluctuations, histone variants, and histone modifications[2–4], all of which play a major role in the regulation of chromatin architecture and genome transactions[5–7]. However, some fundamental aspects of nucleosomes, such as their interplay with DNA topology, remain elusive. In this respect, a paradigmatic case is the so-called "linking number paradox" of nucleosomal DNA[8–10], which has been the subject of debate for decades[11,12].

The linking number ($Lk$) of DNA is the number of times the single strands of the duplex intertwine around each other[12,13]. The $Lk$ paradox refers to the discrepancy between the theoretical and the experimental measurement of the DNA linking number difference ($\Delta Lk$) stabilized by nucleosomes. According to the general equation $\Delta Lk = \Delta Tw + \Delta Wr$[14], it was expected that a nucleosome should stabilize a $\Delta Lk$ value close to $-2$, considering that DNA describes near two left-handed superhelical turns ($\Delta Wr \approx -2$) and assuming no significant changes in the double helical DNA twist ($\Delta Tw \approx 0$). However, numerous studies have persistently concluded that the $\Delta Lk$ constrained by individual nucleosomes is $\sim -1.0$. In those experiments, circular DNA molecules with and without nucleosomes were relaxed with a topoisomerase, and $\Delta Lk$ was calculated. Most of these experiments used the simian virus 40 (SV40) minichromosome as a chromatin model. SV40 was found to have a $\Delta Lk$ of about $-26$[15,16], which was comparable to the number of nucleosomes (24 to 27) typically observed by electron microscopy[17,18]. This $\Delta Lk$ value, which applied to the histone H1-containing native minichromosome, also held true for the H1-free SV40 minichromosome reconstituted in vitro from naked DNA and the four core histones[19]. Another study performed with the yeast circular minichromosome TRP1ARS1 harboring seven nucleosomes also concluded a $\Delta Lk$ value of $-1$ per nucleosome[20]. Finally, in vitro experiments of chromatin reconstitution using tandem repeats of nucleosome positioning sequences and core histones indicated $\Delta Lk$ values of $-1.01 \pm 0.08$[21] and $-1.04 + 0.08$[22] per nucleosome.

The first hypothesis put forward to explain the $Lk$ paradox was that core DNA was notably overtwisted ($\Delta Tw \approx +0.7$)[9,23], which meant that the helical periodicity (h) of DNA was smaller in the nucleosome than in free DNA in the solution (h ≈ 10.5 bp/turn)[24]. The plausible overtwisting of nucleosomal DNA was then calculated by $\Delta Tw = \Delta \emptyset + \Delta STw$, in which the winding number ($\emptyset$) depends on the helical repeat of DNA at the nucleosome surface (hs), and the surface twist ($STw$) is a correction function that accounts for the curved path of DNA in the nucleosome[25]. Multiple measurements of hs based on DNAse I footprinting[23,26–28], hydroxyl radical cleavage[29,30], and DNA base-pair periodicity[31–33] indicated that the mean value of hs is about 10.2 bp/turn, which implied that $\Delta \emptyset \approx +0.4$. The nucleosomal $STw$ was calculated from a derivation for a straight solenoidal helix to be $-0.19$[34,35]. These figures indicated that the overall $\Delta Tw$ of the core DNA is about $+0.2$, a value that was later corroborated by its direct measurement on the nucleosome structure at atomic resolution[1]. The structural data showed also that the core DNA describes about 1.65 left-handed superhelical turns with a pitch angle of about 4 degrees, which produce a $\Delta Wr$ value of about $-1.5$[11,36]. The $Tw$ and $Wr$ deformations of the core DNA were therefore not sufficient to explain the $Lk$ paradox.

A second hypothesis to explain the paradox pointed to the topology of DNA outside the core region. The zig-zag architecture observed in some nucleosomal fibers led to the proposal that if linker DNA segments were repeatedly crossed with a similar geometry, the overall writhe ($\Delta Wr$) of the nucleosomal fiber would increase and produce the apparent $\Delta Lk \approx -1$ per nucleosome[37]. However, recent modeling and experimental measurements with regular arrays of positioned nucleosomes demonstrated that their $\Delta Lk$ varies markedly with nucleosome spacing, such that the apparent $\Delta Lk$ value per nucleosome can range from $-0.8$ to $-1.4$ depending on the DNA linker length[38]. Another proposal involving the topology of DNA outside the core region was based on the study of single nucleosomes reconstituted on small DNA circles[39] and on the torsional resilience of nucleosomal fibers in vitro[40]. These studies suggested that nucleosomes fluctuate between three conformations: one in which incoming and outgoing linker segments form a negative crossing, one with uncrossed linkers, and one in which the linker segments cross positively. As a result, the average $\Delta Wr$ of nucleosomal DNA would be reduced, as would its $\Delta Lk$. However, since these fluctuations depend on external constraints and forces, their plausible relevance to explain the $Lk$ paradox is uncertain.

Here we revisit the $Lk$ paradox of nucleosomal DNA by measuring the $\Delta Lk$ constrained by individual nucleosomes in intracellular chromatin. As a chromatin model, we use small circular minichromosomes of budding yeast, whose nucleosomes are structurally identical to that of higher eukaryotes[41] and are mainly depleted of linker histones[42]. First, we determine the $\Delta Lk$ constrained in minichromosomes containing a known number of nucleosomes. To this end, we fix their DNA topology in vivo and compare it with that of naked DNA relaxed in vitro. We then insert a library of mononucleosomes into these minichromosomes and calculate the $Lk$ gain ($\Delta\Delta Lk$) produced by the individual nucleosomes. Our results indicate that the majority of nucleosomes stabilize $-1.26$ units of $\Delta Lk$. We conclude that this experimental $\Delta Lk$ value, along with the calculated twist and writhe deformations of DNA upon nucleosome formation, provides a solution for the $Lk$ paradox.

## Results

**The $\Delta Lk$ constrained by circular minichromosomes in vivo.** For the purpose of the present study, we constructed YCp1.3, a circular minichromosome of S. cerevisiae comprising only 1341 bp. In order to ensure stable replication and segregation, YCp1.3 contained the genomic TRP1-ARS1 segment and the point centromere CEN2 of yeast (Fig. 1a). The TRP1-ARS1 segment has four nucleosomes (I to IV) occupying the TRP1 coding sequence and a fifth nucleosome (V) positioned 5′ from the ARS1 element[43,44]. CEN2 was then allocated between nucleosomes V and I, upstream of the TRP1 promoter (see Supplementary Fig. 1 for detailed configuration and the bp sequence of YCp1.3). We confirmed the chromatin organization of YCp1.3 via micrococcal nuclease digestion of the minichromosome solubilized from yeast cells. As expected, the DNA sites most sensitive to nuclease digestion occurred in the ARS1 region, followed by the sites corresponding to the linker DNA regions of the point centromere and the five nucleosomes (Fig. 1b).

Next, we determined the $\Delta Lk$ constrained by the chromatin structure of YCp1.3. This value is the difference between the distribution of $Lk$ values of the minichromosome in vivo ($Lk^{CHR}$) and that of the relaxed DNA in vitro ($Lk^0$). To this end, we fixed the $Lk$ values of the minichromosome in vivo by quenching a culture of yeast cells with a freezing ethanol–toluene solution. We showed in previous studies that this quick fixation step irreversibly inactivates cellular topoisomerases, thereby precluding plausible alterations of $Lk^{CHR}$ during cell disruption and DNA extraction[45]. Since $Lk^0$ depends on temperature[46], we relaxed the naked YCp1.3 DNA circle with a type-1B

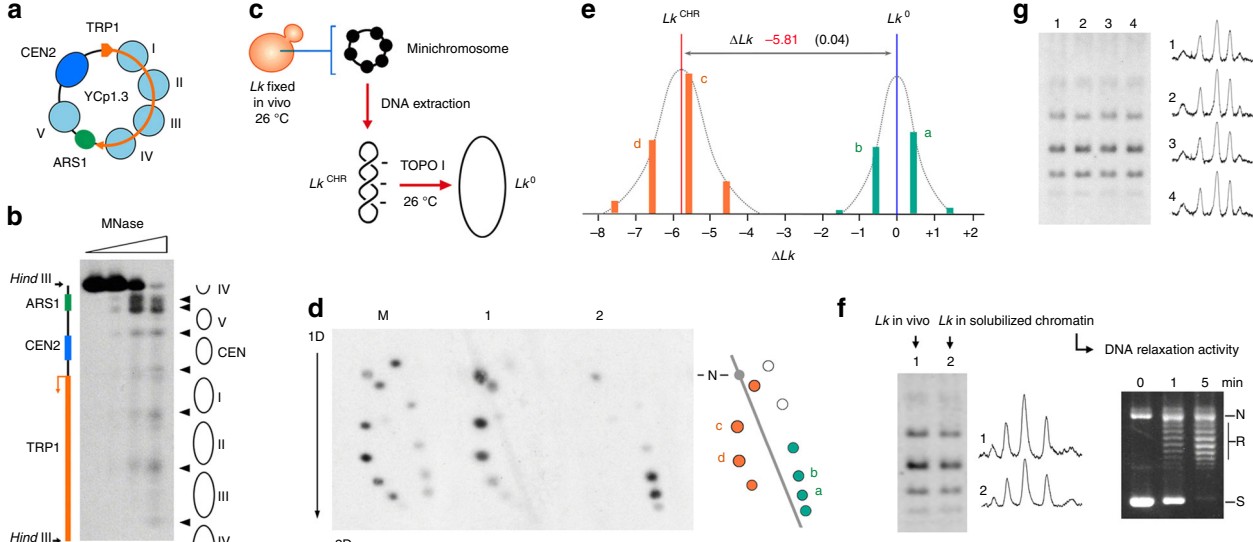

**Fig. 1** Structure and DNA linking number difference in the yeast YCp1.3 minichromosome. **a** Scheme of YCp1.3 (1341 bp) indicating the position of the five nucleosomes (I to V) that occupy the genomic *TRP1-ARS1* segment of *S. cerevisiae*. **b** Micrococcal nuclease digestion pattern of YCp1.3. Nuclease-sensitive sites (arrow heads) are indicated and aligned to functional and structural elements of YCp1.3. **c** Experimental setting to obtain the *Lk* distribution of the YCp1.3 minichromosome in vivo ($Lk^{CHR}$) and the *Lk* distribution of its DNA relaxed in vitro ($Lk^0$). **d** Two-dimensional (2D) electrophoresis of the DNA of the YCp1.3 minichromosome extracted from cells fixed at 26 °C (lane 1) and following relaxation of the naked DNA with topoisomerase I at 26 °C (lane 2). A marker of *Lk* topoisomers of YCp1.3, in which *Lk* values increase clockwise, is included (lane M). DNA electrophoresis, blotting and probing were done as described in the methods. The 2D scheme (right) depicts the relative position of *Lk* topoisomers visible in lane 1 (orange dots) and lane 2 (green dots). Most intense *Lk* topoisomers (a, b, c, d) and nicked (N) molecules are indicated. **e** Intensity plot of *Lk* topoisomers visible in the 2D gel-blot. Colors and letters correspond to those in the 2D scheme. The x-axis indicates $\Delta Lk$ relative to $Lk^0$. The $\Delta Lk$ of the minichromosome (mean ± s.d., $n = 4$) is the difference between $Lk^0$ and $Lk^{CHR}$. See Supplementary Fig. 2 for detailed calculation of $\Delta Lk$. **f** The gel-blot (left) compares the *Lk* distribution of the YCp1.3 minichromosome extracted from fixed cells (lane 1) and that of the YCp1.3 minichromosome solubilized from non-fixed cells (lane 2). Intensity scans of lanes 1 and 2 are shown. The ethidium-stained gel (right) shows DNA relaxation activity in the solubilized chromatin. Supercoiled (S), relaxed (R) and nicked (N) forms of a reporter plasmid are indicated. **g** The gel-blot compares the *Lk* distribution of the YCp1.3 minichromosome in yeast cells cultured in rich medium (lane 1) and synthetic dropout medium (lane 2); and in yeast Δ*top1* (lane 3) and Δ*top1 top2–4* (lane 4) mutants. Intensity scans of lanes 1–4 are shown

topoisomerase at the same temperature (26 °C) used to generate the *Lk* of the YCp1.3 minichromosome in vivo (Fig. 1c). We analyzed the DNA samples using one- or two-dimensional (1D and 2D) agarose gel electrophoresis[47], in which we adjusted the concentrations of chloroquine in order to resolve in the same gel all the *Lk* topoisomers of YCp1.3 in vivo and that of its DNA relaxed in vitro. As seen in the 2D gel in Fig. 1d, the minichromosome (lane 1) and the relaxed DNA (lane 2) presented discrete distributions of *Lk* topoisomers (spots). Such *Lk* distributions occur because the energy difference between the *Lk* topoisomers is less than the thermal energy. The possible *Lk* topoisomers follow a Boltzmann distribution, whose means are $Lk^0$ for the relaxed DNA and $Lk^{CHR}$ for the minichromosome DNA. We subtracted these values and found that YCp1.3 had a $\Delta Lk$ of − 5.81 (Fig. 1e, see Supplementary Fig. 2 for detailed calculation of $\Delta Lk$).

We next asked whether the $\Delta Lk$ value of YCp1.3 was fully constrained by its chromatin structure or could be partially unconstrained (i.e., DNA supercoiling produced by gene transcription). To this end, we examined the topology of YCp1.3 when the minichromosome was solubilized from lysates of unfixed cells (Fig. 1f). In these conditions, cellular topoisomerases present in the cell lysates were able to relax supercoiled DNA plasmids completely (Fig. 1f, right). However, this DNA relaxation activity did not alter the *Lk* distribution of the minichromosome (Fig. 1f, left). We observed also that the *Lk* distribution of YCp1.3 was unchanged when yeast cells were cultured in rich medium and synthetic dropout medium, and

when YCp1.3 was hosted in yeast cells with reduced topoisomerase activity (Δ*top1* and Δ*top1 top2-ts*) (Fig. 1g). All these observations indicated that the $\Delta Lk$ value of − 5.81 was fully constrained by the chromatin structure of YCp1.3. As yeast point centromeres restrain + 0.6 units of $\Delta Lk$[45], the five nucleosomes of YCp1.3 had to stabilize − 6.4 units, an average $\Delta Lk$ of −1.28 per nucleosome.

**Insertion of a mononucleosome library into minichromosomes.** The average $\Delta Lk$ of −1.28 per nucleosome in YCp1.3 assumes that all the minichromosomes are evenly occupied by nucleosomes I to V. However, native yeast nucleosomes are occasionally found to be partially unfolded and invading neighboring nucleosome territories or completely missing[48]. Therefore, the absolute $\Delta Lk$ per nucleosome could be larger than 1.28 if some nucleosomes were missing or unfolded. Conversely, this value could be smaller if the number of assembled nucleosomes were increased, although this possibility is less likely in light of the micrococcal nuclease data and the limited space available in YCp1.3 (Fig. 1a, b). However, we could not discard that chromatin elements other than nucleosomes could also contribute to the $\Delta Lk$ value of YCp1.3. For instance, regulatory factors bound to the *TRP1* promoter and the *ARS1* region may alter the topology of the interacting DNA. Therefore, the average $\Delta Lk$ value of −1.28 per nucleosome estimated above could not be accurate.

To obtain a more reliable $\Delta Lk$ value, we conceived inserting an additional nucleosome into YCp1.3. The difference of $\Delta Lk$ values

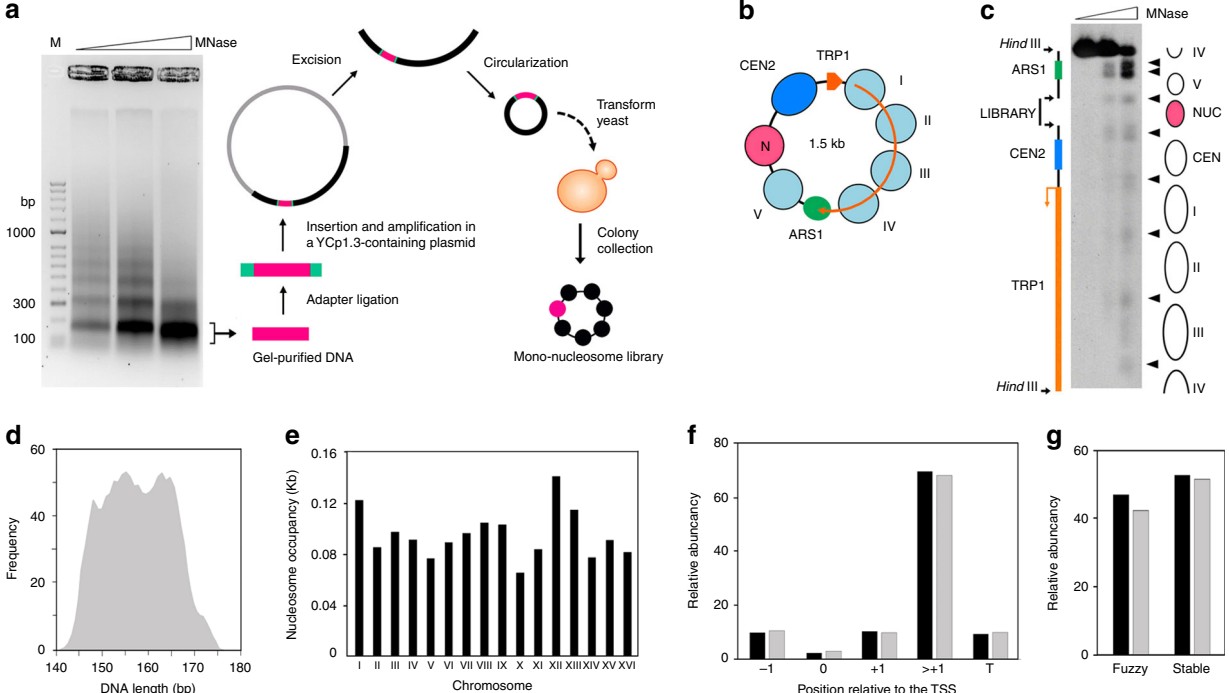

**Fig. 2** Construction and structure of minichromosomes containing a mononucleosomal library. **a** Outline of the procedure used to construct circular minichromosomes containing a nucleosome library. **b** Scheme of the YCp1.3 minichromosome after the inclusion of the nucleosome library (red) between nucleosome V and CEN2. **c** Micrococcal nuclease digestion pattern of the bulk of YCp1.3 minichromosomes containing the library. Structural and functional elements of YCp1.3 and nuclease-sensitive sites (arrow heads) are indicated. **d** Length distribution of nucleosomal DNA fragments inserted in YCp1.3. **e** Chromosomal distribution of the nucleosomal library. The number of nucleosomal DNA sequences normalized per kb is plotted for each of the 16 chromosomes of *S. cerevisiae*. **f** Relative abundance of nucleosome types on the basis of their position relative to the transcription start site (TSS). Following the nomenclature in Jiang and Pugh (2009), "−1" is the first nucleosome upstream of the nucleosome-free region 5′ to TSS, "0" is a nucleosome that overlaps with the nucleosome-free region, "+1" is the first nucleosome downstream of the TSS, and "T" is the terminal nucleosome of a gene. **g** Relative abundance of nucleosomes on the basis of their positional stability. Nucleosomes are classified as "fuzzy or stable", following Ioshikhes et al. (2006). Graphs in **f** and **g** compare relative abundance within our nucleosomal library (black) and within the genomic catalog of nucleosomes (gray)

($\Delta\Delta Lk$) of YCp1.3 with and without this additional nucleosome would indicate the $\Delta Lk$ stabilized by such a nucleosome. The average $\Delta Lk$ constrained by individual nucleosomes in vivo could then be calculated by repeating this experiment with many different nucleosomes. To this end, we constructed a mono-nucleosome library as follows. We digested the whole chromatin of *S. cerevisiae* with increasing amounts of micrococcal nuclease to obtain ladders of nucleosomal DNA fragments. We purified the mononucleosomal DNA fragments (length ≈150 bp), added adapters, and inserted them into the YCp1.3 circle (Fig. 2a). In order not to interfere with the functional elements of YCp1.3, we allocated the nucleosome library insertions between nucleosome V and *CEN2* (Fig. 2b, see Supplementary Fig. 3 for a detailed configuration of the insertion site). Upon transformation of YCp1.3 constructs carrying the mononucleosome library into yeast cells, we collected 1200 colonies. Micrococcal nuclease digestions of the minichromosomes pooled from all the colonies revealed a pattern of DNA cut sites that was nearly identical to that observed in native YCp1.3. However, a new protected DNA segment of about 150 bp appeared between nucleosome V and CEN2, consistent with the expected assembly of an additional nucleosome particle (Fig. 2c).

Parallel sequencing of the full library indicated that nearly all the colonies contained a distinct mononucleosomal DNA fragment inserted between nucleosome V and CEN2 of the YCp1.3 minichromosome. We mapped 1193 different sequences to the reference genome. Their average length was 156 ± 8 bp (mean ± s.d.) (Fig. 2d). We identified them as previously

referenced nucleosomes by intersecting their coordinates with a catalog of nucleosome positions in yeast (Jiang and Pugh, 2009) (Supplementary Data 1). To determine whether our collection of nucleosomes was representative, we examined their chromosomal distribution (Fig. 2e), inter- or intra-genic position relative to transcription start sites[6] (Fig. 2f), and positional stability[49] (Fig. 2g). The relative abundance of these nucleosome classes in our collection was comparable to that of the reference catalog. Therefore, the nucleosome library inserted in the YCp1.3 minichromosome was representative for the purpose of the intended analysis of nucleosomal DNA topology.

**The average $\Delta Lk$ value restrained by individual nucleosomes**. As with YCp1.3, we determined the $\Delta Lk$ of minichromosomes carrying the nucleosome library by comparing their $Lk$ distribution in vivo ($Lk^{CHR}$) with that of their relaxed DNA in vitro ($Lk^0$). Analysis of individual colonies of the library revealed that the minichromosomes had $\Delta Lk$ values in the range of −7.0 to −7.1 (Fig. 3a). Therefore, relative to the $\Delta Lk$ of −5.81 stabilized by YCp1.3, the inserted nucleosomes produced $\Delta\Delta Lk$ of −1.2 to −1.3 units. These values were consistent with the average $\Delta Lk$ of −1.28 per nucleosome calculated for YCp1.3 (Fig. 1e). Moreover, the five clones analyzed in Fig. 3a represented nucleosomes of distinct allocation relative to TSS (−1, +1, >+1) and different positional stability (fuzzy or stable). Therefore, these nucleosomes stabilized comparable $\Delta Lk$ values irrespective of the nucleosome category. The above analysis of individual colonies also showed

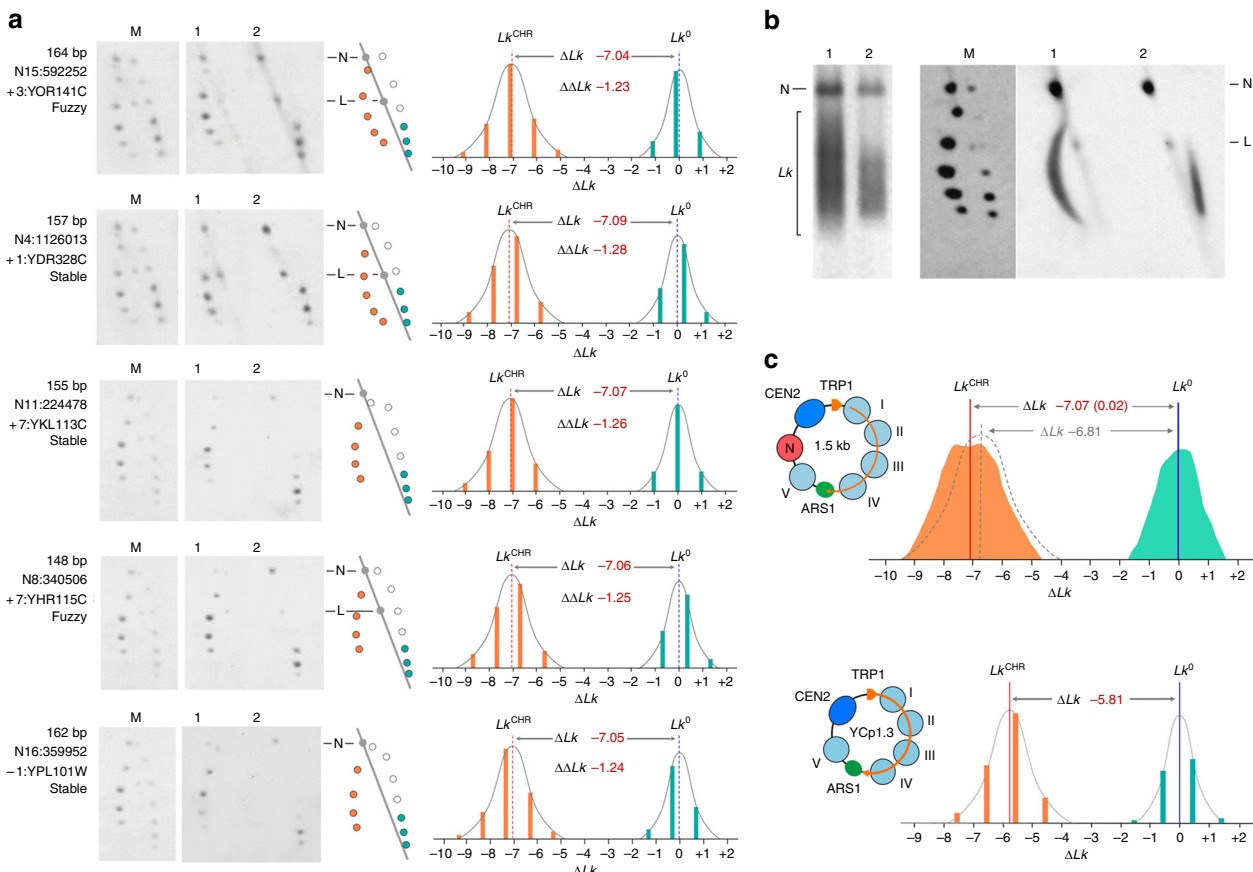

**Fig. 3** $\Delta Lk$ gain produced by the mononucleosome library. **a** DNA topology of five individual minichromosomes of the nucleosomal library. In each case, the length (bp) of the inserted mononucleosomal DNA fragment, the nucleosome ID, genic positioning and stability are indicated. 2D gel-blots show a marker of $Lk$ topoisomers (lane M), the minichromosome DNA extracted from fixed cells (lane 1) and after its relaxation with topo I (lane 2). 2D schemes show the relative position of $Lk$ topoisomers visible in lanes 1 (orange dots) and 2 (green dots). The intensity plots of the above topoisomers showing the mean of the $Lk$ distributions ($Lk^0$ and $Lk^{CHR}$) and the resulting $\Delta Lk$ value were obtained as in Fig. 1e. $\Delta\Delta Lk$ is the $\Delta Lk$ gain produced relative to YCp1.3 ($\Delta Lk = -5.81$). **b** 1D and 2D gel-blots show the DNA of the pool of minichromosomes extracted from the entire library (lane 1) and the pooled DNA after its relaxation with topoisomerase I (lane 2). The 2D gel includes a marker of individual $Lk$ topoisomers (lane M). Nicked (N) and linear (L) molecules are indicated. **c** Top, $Lk$ distributions corresponding to the minichromosomes carrying the nucleosome library (orange) and their DNAs after relaxation with topo I (green). The mean of the pooled $Lk$ distributions ($Lk^0$ and $Lk^{CHR}$) and the resulting $\Delta Lk$ value of $-7.07 \pm 0.02$ (mean ± s.d.) from four replicate experiments were determined as detailed in Supplementary Fig. 4. The position of a hypothetical $Lk$ distribution with $\Delta Lk$ of $-6.81$, which would have implied that the nucleosome library produced a $\Delta\Delta Lk$ of $-1.0$, is illustrated (dashed gray). Bottom, $Lk$ distributions and $\Delta Lk$ of YCp1.3 are shown at the same scale and denote that $\Delta\Delta Lk = -1.26$

that although different minichromosomes had nearly the same $\Delta Lk$, the positions of their $Lk$ topoisomers in the gel did not have the same phasing with respect to $Lk^0$ (Fig. 3a). This misalignment of the $Lk$ topoisomers occurred because the nucleosomal DNA fragments inserted had different lengths (mean 156 bp, s.d. ± 8 bp), and $Lk$ phasing occurs only when the length differences are multiples of the helical repeat of DNA (h ≈ 10.5 bp)[24]. However, the length differences producing $Lk$ misalignment were small compared to the size of the minichromosomes (about 1.57 Kb). Therefore, we were able to analyze the bulk of $Lk$ distributions of the minichromosomes carrying the nucleosome library in a single electrophoresis run, rather than analyzing them individually. To this end, we pooled the colonies of the library to obtain all the $Lk$ distributions of the minichromosomes in one DNA sample and all the $Lk$ distributions of relaxed DNA in another.

As expected, 1D and 2D gel electrophoresis of the pooled samples did not reveal single bands of $Lk$ topoisomers but smeary signals as a result of the overlapping of numerous $Lk$ distributions (Fig. 3b). These overlapped signals presented small protrusions

(Fig. 3c, green and orange), which could suggest that the pooled $Lk$ distributions were bimodal. This outcome would occur if near one half the nucleosomes of the library constrained $\Delta Lk \approx -1.0$ and the other half $\Delta Lk \approx -1.5$. However, this scenario is not consistent with the data of individual nucleosomes, all which constrained $\Delta Lk$ values between $-1.2$ and $-1.3$ (Fig. 3a). Actually, these protuberances were expected for another reason. They appeared because the different lengths of the DNA inserts were not equally represented (Fig. 2d) and thus so was the phasing of their corresponding $LK$ topoisomers. Accordingly, some protrusions appeared also in the pool of relaxed DNAs. The fact that the pooled $Lk$ distributions presented a dispersion similar to that of individual $Lk$ distributions further substantiated that the pooled samples were essentially monomodal. In the case of the relaxed DNA (Fig. 3c, green), the similar dispersion of the pooled and individual $Lk$ distributions corroborated that the gel position of $Lk^0$ was virtually the same for all the DNA molecules of the library regardless of small differences in length. In the case of the minichromosomes (Fig. 3c, orange), the analogous dispersion of the pooled and individual $Lk$

distributions implied that most minichromosomes had nearly the same $Lk^{CHR}$ value. Consequently, most nucleosomes of the library produced the same $\Delta\Delta Lk$ with respect to the $\Delta Lk$ of the YCp1.3 minichromosome. From the means ($Lk^{CHR}$ and $Lk^0$) of the pooled $Lk$ distributions), we found that the minichromosomes containing the nucleosome library had a $\Delta Lk$ of −7.07 (see Supplementary Fig. 4 for detailed calculation). This $\Delta Lk$ value, which represented a $\Delta\Delta Lk$ of −1.26 relative to the $\Delta Lk$ of YCp1.3, was consistent in four replicate experiments (s.d. ± 0.02) and in agreement with that of individual minichromosomes (Fig. 3a). Therefore, we concluded that −1.26 is the average value of the $\Delta Lk$ stabilized by individual nucleosomes. Note that, if the inserted library of nucleosomes had restrained $\Delta\Delta Lk$ values of about −1.0, the minichromosomes would have presented an average $\Delta Lk$ of − 6.81, implying a noticeable displacement of the $Lk^{CHR}$ position in the intensity plot (Fig. 3c).

## Discussion

Here we show experimental evidence that provides a solution to the long-standing $Lk$ paradox of nucleosomal DNA. Our results indicate that most native nucleosomes constrain a $\Delta Lk$ close to −1.26. This value differs markedly from the generally assumed $\Delta Lk$ value of −1.0, which was established in earlier studies. We believe that this discrepancy is due to the distinct chromatin models and limited accuracy of the procedures that were used previously to estimate the $\Delta Lk$ constrained by nucleosome particles.

One source of inaccuracy was in determining the exact number of nucleosomes assembled in circular DNA molecules. Previous studies using SV40 as a chromatin model relied on electron microscopy for counting nucleosomes or nucleosome-like particles. The numbers obtained by different laboratories varied from 20 to 27[17–19,50]. Recent mapping of nucleosome positions in the SV40 genome has revealed that this variability is not only experimental. Intracellular SV40 minichromosomes and SV40 virions present variable nucleosome number and epigenetic modifications that alter the nucleosome organization depending on the infection stage[51]. This variability in nucleosome number could therefore have produced imprecise $\Delta Lk$ values, especially when nucleosome counting and DNA topology analyses were done with uncorrelated samples and by different laboratories[15,16]. Not surprisingly, the SV40 model supported a broad range of $\Delta Lk$ values, including −1.25 per nucleosome[16]. The uncertainty in the exact number of nucleosomes present in circular DNA molecules also affected studies using chromatin reconstitution in vitro, which also relied on electron microscopy for counting nucleosomes[21,22]. Moreover, in these studies, chromatin reconstitution in tandem repeats of nucleosome positioning sequences could have markedly deviated the $\Delta Lk$ values per nucleosome (from −0.8 to −1.4) depending on the periodic length assigned to DNA linker segments[38].

The other source of imprecision in earlier studies was in the calculation of $\Delta Lk$ from the DNA bands observed in agarose gels. In most studies using SV40 and reconstituted chromatin, the gel position of $Lk^0$ was often approximated to that of the slowest $Lk$ topoisomer[15,19,21,22,50], instead of being allocated to the mean $Lk$ of the relaxed $Lk$ distribution[16]. Likewise, in earlier measurements using circular minichromosomes of yeast, the gel position of $Lk^0$ was taken as that of nicked DNA circles. This was the case of the *TRP1ARS1* minichromosome (1.45 kb), which contains seven nucleosomes and was assigned a $\Delta Lk$ of −7[20]. Finally, in most previous studies, it was unclear whether the processing of chromatin samples (to determine $Lk^{CHR}$) and the relaxation of naked DNA circles (to determine $Lk^0$) were quenched at the same temperature. Since the helical repeat of

DNA lessens as the temperature diminishes[46], quenching the topology of DNA at 4 °C produces $Lk$ values up to 1 unit/kb higher than at 37 °C[45].

Our experimental approach minimized the uncertainty in nucleosome counting and $\Delta Lk$ calculation. The small minichromosomes used presented well-defined nucleosome positions, which were bounded by specific chromatin elements (*TRP1* promoter, *ARS1*, *CEN2*). The small size also circumvented significant effects of high order folding of the chromatin on the $\Delta Lk$ values. Our experimental results corroborated the $\Delta Lk$ value stabilized by individual nucleosomes in two ways. First, by averaging the $\Delta Lk$ of the minichromosomes by their number of nucleosomes, we obtained a $\Delta Lk$ of −1.28 per nucleosome. However, this measurement did not take into account plausible variability in nucleosome occupancy and effects of structural elements other than nucleosomes. We reduced these ambiguities by determining the $Lk$ gain ($\Delta\Delta Lk$) produced upon the insertion of the nucleosome library. We obtained thereby the more reliable $\Delta Lk$ value of −1.26 per nucleosome. We found also that there is very little dispersion in the $\Delta Lk$ constrained by the nucleosome library, which indicated that the majority of nucleosomes stabilize a similar DNA topology. Our results could be hardly explained if native nucleosomes were each stabilizing a $\Delta Lk$ of −1.0. This value could stand if the minichromosomes had assembled a number of nucleosomes higher than expected, which seems unlikely in light of the micrococcal nuclease data and the space available. A $\Delta Lk$ of −1.0 per nucleosome could also stand if the minichromosomes were spatially compacted by adopting a strong negative writhe (i.e., $\Delta Wr \approx −1.0$). Such folding would imply that DNA linker lengths and the subsequent rotational orientations between adjacent nucleosomes are alike in all minichromosomes. However, the inserted nucleosome library comprised segments of various lengths and the resulting minichromosomes still constrained very similar $\Delta Lk$ values.

The $\Delta Lk$ value of −1.26 leads to a reevaluation of the $Lk$ paradox of nucleosomal DNA in terms of the general equation $\Delta Lk = \Delta Tw + \Delta Wr$[14]. Considering that the core DNA is globally overtwisted by about + 0.2 turns ($\Delta Tw \approx + 0.2$)[12,35], the stabilization of −1.26 units of $\Delta Lk$ implies that the writhe acquired ($\Delta Wr$) by DNA upon nucleosome formation should be about −1.46. The $Wr$ of DNA in mononucleosomes has been computed to be around −1.5[11,36]. Here we calculated this value for different degrees of superhelical turning of core DNA by $Wr = n(1–\sin\partial)$[14,52], where $n$ is the number of helical turns and $\partial$ is the pitch angle of the turns (supplementary Fig 5). Nucleosomal $Wr$ is about −1.53 when the core DNA completes 1.65 left-handed superhelical turns around the histone octamer. This conformation corresponds to that of the crystallized nucleosome structure[1] and also to that of chromatosomes[53], in which the entry and exit DNA linker segments cross in an angle of about 60° that is fixed by histone H1. This $Wr$ value is likely to reflect thus the upper limit of the absolute DNA writhe of nucleosomes in solution. However, yeast has very low level of linker histone[54], though nuclease digestions support the existence of proto-chromatosome structures[55]. Moreover, extensive experimental evidence has demonstrated that the conformational dynamics of nucleosomes in physiological conditions frequently leads to partial unwrapping or breathing motions of the core DNA[56–59]. These motions substantially reduce the absolute $Wr$ of the nucleosomal DNA and thereby its average value. For instance, just by reducing the wrapping of DNA to 1.5 left-handed superhelical turns, mononucleosomal $Wr$ drops to −1.38 (supplementary Fig 5). Therefore, an average $\Delta Wr$ of about −1.46 per nucleosome is a realistic topological mark, which along the $\Delta Tw$ of about + 0.2 and the $\Delta Lk$ of −1.26, provides a plausible explanation for the linking number paradox of nucleosomal DNA.

Our experimental findings contribute to a better understanding of how DNA supercoiling energy is confined by nucleosomes, and of how nucleosomes buffer the DNA supercoiling generated during gene transcription. Our experimental approach also leads to a new genome-wide categorization of nucleosomes on the basis of their DNA topology, thus opening a new dimension toward deciphering the mechanisms that orchestrate chromatin structure and functions.

## Methods

**Construction of minichromosomes and the nucleosome library.** To construct YCp1.3 (1341 bp), a 997 bp genomic segment of *S. cerevisiae* containing *TRP1-ARS1* (coordinates 461739 to 462736) and a 243 bp genomic segment containing *CEN2* (coordinates 238194 to 238437) were amplified by PCR. Both segments were ligated and inserted into a plasmid vector via endonuclease restriction sites engineered by PCR. Subsequent digestion with endonuclease *NotI* released a 1341-bp fragment containing the *TRP1-ARS1-CEN2* sequence. This fragment was circularized with T4 DNA ligase and monomeric circles were gel-purified to obtain the YCp1.3 circle. See Figure S1 for a description of the oligonucleotides used for PCR and the complete bp sequence of YCp1.3. The YCp1.3 circle was used to transform the *S. cerevisiae* strain FY251 (*MATa his3-Δ200 leu2-D1 trp1-Δ63 ura3–52*) and its topoisomerase-mutant derivatives JCW27 (*Δtop1*) and JCW28 (*Δtop1 top2–4*)[60]. To construct the nucleosome library, yeast cells from a 250 ml culture (OD 1.0) were collected, washed with water, and incubated with 80 ml of 1 M Sorbitol, 30 mM DTT for 15 min at 28 °C. Next, 625 U of Lyticase (Sigma-Aldrich L2524) and 10 μL of 4 M NaOH were added to the cells suspension, and the incubation continued until > 80% of cells were converted into spheroplasts. Spheroplasts were washed with 1M Sorbitol and resuspended in 1.5 ml of hypotonic lysis buffer (1 mM CaCl₂ 5 mM KH₂PO₄ 1 mM PMSF) at 24 °C. Next, 30 units of micrococcal nuclease (Sigma-Aldrich N3755) were added, and the mixture was incubated at 24 °C. Aliquots of 300 μl were quenched with 20 mM EDTA 1% SDS at different time points (3–30 min). The digestion of chromosomal DNA was examined by gel electrophoresis (1% agarose in TBE buffer, 80 V during 3 h). Mononucleosomal DNA fragments (about 150 bp in length) produced at different time points were gel-eluted and pooled. The severed DNA fragments produced by micrococcal nuclease were repaired by removing terminal 3′-phosphates with T4-polynucleotide kinase and by filling the DNA ends with Klenow and T4-DNA polymerase activities. The nucleosomal DNA fragments were A-tailed with Klenow activity and ligated to adapters. The adapters included an *AscI* site, which permitted the insertion of the mononucleosomal DNA fragments into the single *AscI* site of YCp1.3. See Figure S2 for a description of the adapters and the site of insertion in YCp1.3. The YCp1.3 constructs containing the library of mononucleosomal DNA sequences were amplified in bacterial plasmids. Upon digestion with endonuclease *NotI*, the fragments of about 1.57 Kb containing the library within YCp1.3 were circularized with T4 DNA ligase. Monomeric circles were gel-purified and used to transform FY251.

**Yeast culture and DNA extraction of fixed minichromosomes.** Yeast cells transformed with YCp1.3 were grown at 26 °C in standard yeast synthetic media containing TRP dropout supplement (Sigma) with 2% glucose or in rich YPD medium, as indicated. When the liquid cultures (20 ml) reached mid-log phase (OD ≈ 0.8), the cells were fixed in vivo by quickly mixing the cultures with one cold volume (−20 °C) of ET solution (Ethanol 95%, Toluene 28 mM, Tris-HCl pH 8.8 20 mM, EDTA 5 mM). The following steps were done at room temperature. Cells were sedimented, washed twice with water, resuspended in 400 μl of TE, and transferred to a 1.5-ml microfuge tube containing 400 μl of phenol and 400 μl of acid-washed glass beads (425–600 μm, Sigma). Mechanic lysis of >80% cells was achieved by shaking the tubes in a FastPrep® apparatus for 10 s at power 5. The aqueous phase of the lysed cell suspension was collected, extracted with chloroform, precipitated with ethanol, and resuspended in 100 μl of TE containing RNAse-A. After a 15-min incubation at 37 °C, the samples were extracted with phenol and chloroform, DNA precipitated with ethanol and resuspended in 30 μl of TE. The same procedure of cell culture, in vivo fixation, cell lysis and DNA extraction was applied to individual colonies of the minichromosome library. In the case of sampling the full library, the bulk of colonies were collected from agar plates, washed with water, and diluted (OD 0.2) in 200 ml of standard yeast synthetic media (TRP dropout). The pooled cells were grown at 26 °C and fixed when the liquid cultures reached mid-log phase.

**Yeast culture and solubilization of minichromosomes.** Liquid yeast cultures (20 ml) at mid-log phase were sedimented, washed with water, and resuspended in 500 μl of buffer L (Tris-HCl 10 mM pH 8, EDTA 1 mM, EGTA 1 mM, DTT 1 mM, NaCl 150 mM, Tritón 0.1%, pepstatin 1 mg/ml, leupeptin 1 mg/ml, and PMSF 1 mM). The suspension was transferred to a 1.5-ml microfuge tube containing 500 μl of acid-washed glass beads (425–600 μm, Sigma). Mechanical lysis of >80% cells was achieved after six cycles of 30 s of vortexing plus 30 s of ice cooling. The supernatant of the lysate was recovered by centrifugation (2000 × *g*) and loaded on a Sephacryl S-300 column equilibrated with buffer L at 4 °C. The first filtration volume containing the circular minichromosomes was recovered and incubated at 26 °C for 10 min. A supercoiled DNA plasmid was added to one aliquot of the eluted minichromosomes and further incubated for 5 min in order to test the DNA relaxation activity of cellular topoisomerases. The reactions were then quenched with one volume of 20 mM EDTA, 1% SDS, and 100 mg/ml proteinase K, followed by an incubation at 60 °C for 30 min. The samples were extracted with phenol and chloroform, and the DNA was precipitated with ethanol and resuspended in 30 μl of TE.

**Micrococcal nuclease mapping of chromatin structure.** YCp1.3 minichromosomes and their derivatives were solubilized and eluted from a Sephacryl S-300 column as described above. Eluted minichromosomes were adjusted to 2 mM CaCl₂ and pre-incubated at 25 °C for 5 min. Micrococcal nuclease was added (2–100 units/ml), and incubations proceeded at 25 °C for 5 min. The digestions were quenched with one volume of 20 mM EDTA, 0.5% SDS, and 100 mg/ml proteinase K, followed by incubation at 60 °C for 30 min. The mixtures were extracted with phenol and chloroform, and the DNA precipitated with ethanol and resuspended in 30 μl of TE. The DNA was then digested with *HindIII* restriction endonuclease, which has a single cutting site in YCp1.3. The resulting DNA fragments were separated by gel electrophoresis (1.2% agarose), blotted, and probed with a short DNA sequence (194 bp) starting at the *HindIII* site of YCp1.3.

**DNA sequencing and analysis.** DNA extracted from minichromosomes containing the mononucleosome library was sequenced (Illumina HiSeq 2000, 50 base paired-end reads), and resulting FASTQ data files were subject to QC using Cutadapt (1.12). Sequences were then mapped to the *S. cerevisiae* reference genome (SacCer3) using bowtie (v1.1.2). Once nucleosome coordinates had been established, subsequent analyses were performed by integrating published data sets (Ioshikhes et al 2006; Jiang and Pugh, 2009) and by using bedtools (v2.27) and Galaxy.

**DNA relaxation with topoisomerase I.** DNA purified from minichromosome preparations was pre-incubated at 26 °C for 5 min in 30 μl of Tris-HCl 10 mM pH 8, EDTA 1 mM, and NaCl 150 mM. Catalytic amounts of topoisomerase I of vaccinia virus[45] were then added, and the incubations proceeded at 26 °C for 30 min. Reactions were quenched with one volume of 20 mM EDTA and 1% SDS.

**Electrophoresis of *Lk* topoisomers.** DNA from YCp1.3 (1341 bp) and from minichromosomes containing the mononucleosome library (about 1.57 kb) were loaded onto 1.4% (w/v) agarose gels. One-dimensional electrophoresis was carried out at 2.5 V/cm for 18 h in TBE buffer (89 mM Tris-borate and 2 mM EDTA) containing 0.55 μg/ml chloroquine. Two-dimensional electrophoresis was in TBE containing 0.55 μg/ml chloroquine in the first dimension (2.5 V/cm for 18 h) and in TBE containing 3 μg/ml chloroquine in the second dimension (5 V/cm for 4 h). Gels were blot-transferred to a nylon membrane and probed at 60 °C with the *TRP1ARS1* sequence labeled with A*Lk*Phos Direct (GE Healthcare®). Chemiluminescent signals of increasing exposure periods were recorded on X-ray films and non-saturated signals of individual *Lk* topoisomers and bins of pooled *Lk* distributions quantified with ImageJ.

**Lk distribution analysis and calculation of ΔLk.** In the case of individual minichromosomes, the most intense topoisomer of the *Lk* distribution of relaxed DNA was initially assigned the value Δ*Lk* = 0. An Integer Δ*Lk* value (positive or negative) was subsequently assigned to each topoisomer of the *Lk* distributions of minichromosome and relaxed DNA according to the *Lk* markers included in the 2D gels. The mean value of each *Lk* distribution (*Lk*⁰ and *Lk*^CHR) was calculated, and the Δ*Lk* between the *Lk* distributions was obtained by subtracting their means (see details in Supplementary Fig. 2). In the case of pooled minichromosomes, continuous *Lk* distributions were quantified by bins and their mean calculated. The gel position of the means was interpolated with the that of *Lk* marker to obtain Δ*Lk* values of the means in the marker frame. The Δ*Lk* between pooled minichromosomes and relaxed DNAs was determined by subtracting their means (see details in Supplementary Fig. 4). In all figures, the *Lk* results were illustrated by plotting the intensity of *Lk* topoisomers of minichromosomes and relaxed DNA along a scale of Δ*Lk* units (*x*-axis), in which the value Δ*Lk* = 0 was re-adjusted to the mean of the *Lk* distribution of the relaxed DNA (*Lk*⁰).

## Data availability

The data that support the findings of this study are available from the corresponding author upon request.

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

## Acknowledgements

This work was supported by grants from Plan Estatal de Investigación Científica y Técnica of Spain to J.R. (BFU2015-67007-P and MDM-2014-0435-02). J.R. is a Professor at the Spanish National Research Council (CSIC).

## Author contributions

J.R. conceived the project. J.R. and J.S. designed experiments. J.S., O.D., B.M., A.V. and S.D. performed experiments. J.R., J.S. and R.S.J. analyzed data. J.R. wrote the manuscript.

## Additional information

**Competing interests:** The authors declare no competing interests.

