## [Peer Review File · Nature Communications]

Reviewer #1 (Remarks to the Author):

The two strands of a circular DNA plasmid possess a topological invariant, called the linking number (Lk), which is equal to the number of times the single strands of DNA wind around each other. An unstressed B-DNA molecule contains a right-handed twist every ~ 10.5 bp, corresponding to a length $h = 3.4$ nm, so a planar unstressed circular DNA molecule of length L has a linking number $Lk_0 = L / h$. Changes in linking number, $\Delta Lk = Lk - Lk_0$, characterize the degree of DNA supercoiling.

Two-dimensional (2D) agarose gel electrophoresis using intercalating agents is the usual way of distinguishing topological isomers (or topoisomers) - species of DNA molecules characterized by different Lk (different degrees of supercoiling). Using a library of mononucleosomes inserted into small plasmids, the authors show that nucleosomes stabilize a change in Lk of about $\Delta Lk = -1.26$, and claim to resolve the "linking number paradox", i.e. different experimental measurements do not agree on the equality $Lk = Tw + Wr$, where Tw and Wr are the two geometrical quantities (twist and writhe) - see e.g. Marko, Siggia - Phys Rev E 1995 for a rigorous definition of these DNA properties.

I have found a few places in this manuscript that were a bit confusing to me, and I believe this manuscript would significantly improve if the authors could clarify the following points:

1. For the non-experts in this field, the authors should briefly explain why the resulted 2D gels contain multiple topoisomer spots. Does this mean that different plasmids contain different numbers of nucleosomes, or that topoisomerases do not remove the same amount of twist from each plasmid, or something else?

2. The Lk_0 distribution from Fig. 1E, which quantifies the intensities of the topoisomer spots from the 2D gel in Fig. 1D, contains only 2 values and approximating this distribution (2 green bars) with a Gaussian is not clearly justified in the text. Both distributions shown in Fig. 1E are discrete distributions (they contain discrete values), so I don't understand why they are fitted using Gaussian distributions - this should be clearly explained in the text, as it is not obvious. Gaussians are continuous distributions so using them to approximate a discrete distribution seems odd. Moreover, I don't think it is even necessary to fit these distribution with any model in order to obtain the distribution of ΔLk . If the two distributions for the values of Lk and Lk_0 , $\text{Prob}(Lk = x)$ and $\text{Prob}(Lk_0 = x)$, are available, then one can easily compute the probability distribution for every value of ΔLk : $\text{Prob}(\Delta Lk = N) = \sum \text{over } x \text{ of } \text{Prob}(Lk = N+x) \text{ Prob}(Lk_0 = x)$.

3. Fig. 1E and throughout the main text: It is not clear what the authors mean by "the midpoint/center of the distribution" - that is not a rigorous mathematical term to characterize a

probability distribution. Is that the mean, the median, the most probable topoisomer in the distribution, or the mean/maximum of the fitted Gaussian? The same question for Fig. 3C: "The center of intensity of the pooled Lk distributions..." - what exactly is meant by this?

4. Fig. 1D could benefit from some extra annotations to indicate what those bands/spots represent. For example, the authors could number the spots representing various topoisomers, and indicate the direction of decreasing Lk (which may not be obvious for readers that are not familiar with the 2D gel electrophoresis technique). That way one could easily relate the topoisomers from the gel with the corresponding bars from the intensity plot (Fig. 1E).

5. Why aren't the bars in Fig. 1E located at integer values of Lk, and how were these values computed? The highest green bar corresponds to the most abundant topoisomer from lane 2, and usually this is denoted as the "0" topoisomer and all other topoisomers are numbered/counted using this as a reference.

6. Also regarding Fig. 1E, while the numbering of the green bars is evident (the highest bar is typically denoted by "0"), how exactly are the yellow bars (topoisomers from lane 1) numbered/counted, since multiple intermediary topoisomer spots are missing from the 2D gel (white circles in the cartoon). In other words how can one decide whether the darkest topoisomer spot from lane 1 corresponds to topoisomer "-5", "-6", or some other topoisomer, when it is not clear how many topoisomers are missing between the "0" topoisomer and the specific topoisomer that needs to be labeled. I think a "marker" lane should be included in all the 2D gels in order to make the topoisomer numbering unambiguous. I think it is very important to clarify this aspect since all the results depend on the number of missing spots, and the authors use different numbers in different figures, e.g. 3 missing spots in Fig. 1D, and 3 or 4 missing spots in Fig. 3A. So, why using 3 in one figure and 4 in another?

7. Related to my previous question, how are these "topoisomer numbers" (values on the x axis) estimated in Fig. 3C (top panel), as in the case of the pooled library it is impossible to count/number distinct topoisomers (see lanes 2,3 in Fig. 3B)?

8. Why are the x values in Fig. 3A different in the 4 experiments that are shown? The highest green bars should correspond to the "0 topoisomer" in all 4 experiments and all other topoisomers should have similar integer linking numbers. The axes in these plots should be clearly labeled - both x and y labels are missing here.

9. The x label of Fig. 1E, "Lk units", is a bit confusing. I suppose this axis actually indicates $\Delta Lk = Lk - Lk_0$, where Lk_0 is the linking number of the relaxed circle ($Lk_0 = L/h$) and Lk is the linking number of the given topoisomer. Please clarify what "Lk units" means, since Lk is a dimensionless quantity, so it doesn't have units of measurement - which makes "Lk units" confusing. Maybe a better label would be "Topoisomer spot (ΔLk)"?

10. Please explain in more details how exactly ΔLk was computed. It seems to me that there are different ways to estimate this, e.g. more rigorously by computing its probability distribution as explained at point 2, or more arbitrarily by subtracting different quantities, e.g. the means of the 2 distributions, the medians of the 2 distributions, the two most abundant topoisomers, or the fitted Gaussian means - which seems to be the preferred way of the authors. I don't understand why this is the correct way to estimate ΔLk , and consequently I can't appreciate the significance of the obtained result $\Delta Lk/\text{nucleosome} = -1.26$

11. In all the estimations, the authors assume that a fixed number of nucleosomes are found on every plasmid. For example from Fig. 1 it is obtained that $\Delta Lk = -5.8$, and it is assumed that all plasmids will contain 6 nucleosomes (including 1 centromeric nucleosome). I believe nucleosomes are more dynamic than assumed here, and some plasmids may contain 6 nucleosomes while other only 5 nucleosomes. So how can one eliminate this possibility, which could also generate $\Delta Lk = -5.8$: each nucleosome stabilizes $\Delta Lk/\text{nucleosome} = -1$ but some plasmids contain only 5 nucleosomes, and most of them contain 6 nucleosomes, giving an average of $\Delta Lk/\text{plasmid} = -5.8$?

Reviewer #2 (Remarks to the Author):

The authors reexamine a fundamental question in chromatin biology: what is the linking number of DNA containing a nucleosome. Using their published quenching method of capturing plasmids at their in vivo supercoiling state, they determine the linking number for a single nucleosome in their plasmid as -1.28. They further confirm this number by adding an additional nucleosome containing sequence generated from MNase digest of yeast chromatin. Both with individual clones from the genomic inserts and by pooling all clones, they arrive at similar linking number around -1.26. They then recalculate the writhe from a nucleosome structure and arrive at -1.53. So the combination of this writhe value and overtwisting of DNA on nucleosome surface (+0.2) leads to a theoretical expectation of -1.3 which is very close to the experimental values obtained by them (-1.26), thus resolving the linking number paradox.

This paper presents well designed experiments answering a long-standing question. The paper is also written well, putting the new results in the proper historical context. I have some minor comments the authors need to address:

1. The authors need to address in the introduction if the writhe for nucleosome has been determined before, and if not that this also needs to be accounted for to obtain a theoretical linking number change. For example in the introduction, the authors state “the superhelical turning of DNA around a histone octamer ($\Delta W_r \approx -1.7$) predicts that a nucleosome should stabilize about -1.7 units of ΔLk ”, but in discussion, they say that writhe has to be calculated. They need to reconcile these two statements in the introduction itself.

2. In figure 3C, the orange curve seems to be a bimodal distribution, implying a population of nucleosome sequences may have $\Delta Lk > -1.26$. The authors should fit a bimodal distribution to the orange curve and determine the two mean positions and the resulting two values of linking number changes for the population.

3. In Figure 3A, four representative clones are shown. The authors should indicate which nucleosome position these clones belong to and if they are fuzzy vs. stable. If all four representative clones are similar (for example all are gene body nucleosomes), they should show 2D gels for representative +1, 0, -1 and fuzzy nucleosomes.

Reviewer #3 (Remarks to the Author):

This manuscript addresses the long-known discrepancy between the nucleosome core structure known from X-ray crystal studies and DNA topology measurements in native covalently closed circular minichromosomes known as the “linking number paradox”. Initially, studies of SV40 virus minichromosomes containing chains of 22-26 nucleosomes showed that the observed nucleosome-imposed changes in the topological invariant (ΔLk) of about -1 per nucleosome are substantially less than -1.7 predicted from the known nucleosome core structure. Subsequently, studies in biochemically defined in vitro reconstituted mini-circles showed that this difference is mostly due to the linker DNA length and conformation. However, the question of actual ΔLk values generated by native minichromosomes of various linker DNA lengths in vivo remains open. In this work, Segura et al. designed a small yeast minichromosome system in which the number of nucleosome positions can be easily determined and quantified. Importantly, they inserted a library of randomly selected yeast mononucleosomes in their construct that allowed them to quantitate a relative gain of ΔLk per one added nucleosome. In this system, they observed a value of $\Delta Lk = -1.26$ per nucleosome which, in addition to the realistic assumptions for DNA double helical twist ($\Delta Tw = +0.2$) and superhelicity

($\Delta W_r = -1.46$) predicted for slightly unfolded mononucleosomes is closer to the ΔL_k predicted from the nucleosome core X-ray crystal structure than the earlier SV40 studies. This is an important and novel experimental observation that contributes to understanding DNA topology and conformation in native chromatin.

Overall, this is a high-quality experimental work, which introduces an innovative minichromosome system to study nucleosome DNA topology and reports interesting and potentially important experimental findings. The manuscript is concise and well written. However, I have a significant concern that important structural parameters describing state of DNA in yeast minichromosomes such as W_r , T_w , and nucleosome occupancy are not experimentally determined but rather subjectively selected from literature to match the ΔL_k expected to solve the so-called superhelical paradox. In view of the above concern, the primary conclusion about cracking the linking number paradox remains speculative. To ensure its publication, the title and the main conclusions of the manuscript should be changed to properly reflect the data and/or substantial additional experiments should be conducted to determine the actual structure and nucleosome occupancy of the minichromosomes.

Specific points:

1. My strongest concern is that the actual number of nucleosomes in the minichromosomes should be precisely determined in order to deduce the number of ΔL_k per nucleosome. Current MNase data provide decent measurements of nucleosome positioning but not occupancy and structural variations. Native yeast nucleosomes are notorious for being variously unfolded and invading neighboring nucleosome territories, or completely missing (see e.g. Chereji and Morozov, 2014, PNAS 111, 5236-5241) potentially imposing strong and unknown effects on DNA superhelicity. In classic experiments with isolated SV40 minichromosomes, EM observations were crucial in determining the number of nucleosomes and the beads on a string structure of the minichromosome. Here, to corroborate conclusions of this paper, independent measurements of nucleosome occupancy on individual minichromosome should be conducted if possible by some single minichromosome imaging technique (TEM, AFM, super-resolution fluorescence imaging), otherwise measuring of specific ΔL_k value per one nucleosome is likely incorrect.
2. YCp1.3 is a transcriptionally active minichromosome with promoter that may contain transcriptional factors and RNA polymerase II that generate torsional tension. This dynamic torsional tension may affect nucleosome structure making nucleosome chains to accumulate additional negative superhelicity distributed to all nucleosomes (including the added library nucleosomes). The effect of transcription should be addressed by experiments in which either RNApolIII or topoisomerases are inhibited before fixing the nucleosomes or the added nucleosome insulated from the transcription effects.

3. ΔW_r prediction per nucleosome (Fig S3) is based on a isolated mononucleosome and does not take into account the linker DNA length and conformation that are likely to change W_r considerably in a nucleosome array as discussed by the authors in the introduction. Modeling of a circle nucleosome chain rather than a single nucleosome is essential for correct prediction of ΔW_r .

4. The authors arbitrary consider the effect of partial nucleosome unwrapping but not that of the increased wrapping of linker DNA that resembles chromatosome and is observed in yeast despite the very low level of linker histone (Cole et al., 2016, NAR 44, 573-581). Precise nucleosome mapping with single nucleotide resolution may help in deducing the linker DNA length and the extent of DNA unwrapping/overwrapping.

5. Abstract, 9th line should read ($\Delta T_w \approx +0.2$), not ($\Delta T_w \approx +2.0$)

Point-by-point response to the referees' comments

We are very pleased with all the suggestions and questions raised by the 3 reviewers. Their comments have been very constructive allowing us to improve the clarity and strength of the study. All the points raised are addressed in the revised text (changes in red) and in the point-by-point responses (italicized).

Reviewer #1

The two strands of a circular DNA plasmid possess a topological invariant, called the linking number (Lk), which is equal to the number of times the single strands of DNA wind around each other. An unstressed B-DNA molecule contains a right-handed twist every ~ 10.5 bp, corresponding to a length $h = 3.4$ nm, so a planar unstressed circular DNA molecule of length L has a linking number $Lk_0 = L/h$. Changes in linking number, $\Delta Lk = Lk - Lk_0$, characterize the degree of DNA supercoiling.

Two-dimensional (2D) agarose gel electrophoresis using intercalating agents is the usual way of distinguishing topological isomers (or topoisomers) - species of DNA molecules characterized by different Lk (different degrees of supercoiling). Using a library of mononucleosomes inserted into small plasmids, the authors show that nucleosomes stabilize a change in Lk of about $\Delta Lk = -1.26$, and claim to resolve the "linking number paradox", i.e. different experimental measurements do not agree on the equality $Lk = Tw + Wr$, where Tw and Wr are the two geometrical quantities (twist and writhe) - see e.g. Marko, Siggia - Phys Rev E 1995 for a rigorous definition of these DNA properties.

I have found a few places in this manuscript that were a bit confusing to me, and I believe this manuscript would significantly improve if the authors could clarify the following points:

Most questions of the reviewer relate to the analysis and graphical representation of the Lk data (points 2 to 10). We realize that the explanation of these aspects was too concise in the initial submission. Therefore, we have added a new methods subsection ("Lk distribution analysis and calculation of ΔLk ") and a new figure (supplementary Fig 2) that clarify these matters.

1. For the non-experts in this field, the authors should briefly explain why the resulted 2D gels contain multiple topoisomer spots. Does this mean that different plasmids contain different numbers of nucleosomes, or that topoisomerases do not remove the same amount of twist from each plasmid, or something else?

As requested, we clarified this point in the revised results (subsection 1, paragraph 2):

..."As seen in the 2D gel in Fig. 1d, the minichromosome (lane 1) and the relaxed DNA (lane 2) presented discrete distributions of Lk topoisomers (spots). Such Lk distributions occur because the energy difference between the Lk topoisomers is less than the thermal energy. Therefore, Lk distributions tend to be Gaussian around a mean value, which is Lk^0 for the relaxed DNA and Lk^{CHR} for the minichromosome DNA"....

2. The Lk_0 distribution from Fig. 1E, which quantifies the intensities of the topoisomer spots from the 2D gel in Fig. 1D, contains only 2 values and approximating this distribution (2 green bars) with a Gaussian is not clearly justified in the text. Both distributions shown in Fig. 1E are discrete distributions (they contain discrete values), so I don't understand why they are fitted using Gaussian distributions - this should be clearly explained in the text, as it is not obvious. Gaussians are continuous distributions so using them to approximate a discrete distribution seems odd. Moreover, I don't think it is even necessary to fit these distributions with any model in order to obtain the distribution of ΔLk . If the two distributions for the values of Lk and Lk_0 , $\text{Prob}(Lk = x)$ and $\text{Prob}(Lk_0 = x)$, are available, then one can easily compute the probability distribution for every value of ΔLk : $\text{Prob}(\Delta Lk = N) = \sum \text{over } x \text{ of } \text{Prob}(Lk = N+x) \text{Prob}(Lk_0 = x)$.

The reviewer is right in that approximating discrete Lk values to Gaussian distributions seems odd. However, it is well-established that Lk distributions are Gaussian (as explained in point 1). This is evident in larger DNA rings (5-10 kb), where Lk distributions contain numerous topoisomers. As topoisomers can only have integer Lk values, small DNA rings (<1.5 kb) present few Lk topoisomers in their distributions. We also agree in that it is not necessary to fit the distributions with any model in order to obtain ΔLk . In this respect, we highlight in the revision (new methods subsection and new supplementary fig 2) that we do not use this fitting to calculate ΔLk . Namely, we determine the mean of each Lk distribution from the topoisomer intensities, and then subtract the means to obtain ΔLk . Only afterwards, we fit bell-shapes just for illustrative purposes.

3. Fig. 1E and throughout the main text: It is not clear what the authors mean by "the midpoint/center of the distribution" - that is not a rigorous mathematical term to characterize a probability distribution. Is that the mean, the median, the most probable topoisomer in the distribution, or the mean/maximum of the fitted Gaussian? The same question for Fig. 3C: "The center of intensity of the pooled Lk distributions..." - what exactly is meant by this?

Certainly, the terms "the midpoint/center of the distribution/center of intensity" are ambiguous. Therefore, we removed them. As indicated throughout the revision (results, methods and supplementary fig 2), we used the mean (Lk^0 and Lk^{CHR}) to characterize the probability distributions.

4. Fig. 1D could benefit from some extra annotations to indicate what those bands/spots represent. For example, the authors could number the spots representing various topoisomers, and indicate the direction of decreasing Lk (which may not be obvious for readers that are not familiar with the 2D gel electrophoresis technique). That way one could easily relate the topoisomers from the gel with the corresponding bars from the intensity plot (Fig. 1E).

We followed the reviewer's suggestions to facilitate the identification of 2D gels spots and their corresponding bars in the intensity plot. The new Supplementary Fig 2 explains in detail how this was done for fig 1d. We modified also Fig 1, by adding a Lk marker in the 2D gel (indicating that Lk increases clockwise) and labelling intense Lk topoisomers in lanes 1 and 2 (a, b, c, d) both in the gel and the plot.

5. Why aren't the bars in Fig. 1E located at integer values of Lk, and how were these values computed? The highest green bar corresponds to the most abundant topoisomer from lane 2, and usually this is denoted as the "0" topoisomer and all other topoisomers are numbered/counted using this as a reference.

This matter is also clarified in the new methods subsection and new supplementary fig 2. Namely, we start by pre-assigning integer Lk values to the topoisomers, being the most intense topoisomer of the relaxed distribution denoted as the "0" (as the reviewer specifies). However, once we have the means and ΔLk results, we adjust the value ΔLk "0" to Lk^0 . For this reason, the Lk topoisomers do not have integer values in the x-axis (ΔLk) of the plots.

6. Also regarding Fig. 1E, while the numbering of the green bars is evident (the highest bar is typically denoted by "0"), how exactly are the yellow bars (topoisomers from lane 1) numbered/counted, since multiple intermediary topoisomer spots are missing from the 2D gel (white circles in the cartoon). In other words, how can one decide whether the darkest topoisomer spot from lane 1 corresponds to topoisomer "-5", "-6", or some other topoisomer, when it is not clear how many topoisomers are missing between the "0" topoisomer and the specific topoisomer that needs to be labeled. I think a "marker" lane should be included in all the 2D gels in order to make the topoisomer numbering unambiguous. I think it is very important to clarify this aspect since all the results depend on the number of missing spots, and the authors use different numbers in different figures, e.g. 3 missing spots in Fig. 1D, and 3 or 4 missing spots in Fig. 3A. So, why using 3 in one figure and 4 in another?

As requested, we included a "marker" (lane M) in all the 2D figures in order to make the topoisomer numbering unambiguous. Supplementary Fig 2 further explains topoisomer counting.

7. Related to my previous question, how are these "topoisomer numbers" (values on the x axis) estimated in Fig. 3C (top panel), as in the case of the pooled library it is impossible to count/number distinct topoisomers (see lanes 2,3 in Fig. 3B)?

Certainly, it is not possible to count topoisomers in the pooled distributions (lanes 2,3 in Fig. 3b). We used the Lk marker in lane M to compute the Lk difference (ΔLk) between the mean intensity of the pooled distributions of relaxed DNA (Lk^0) and that of the pooled distributions of minichromosome DNA (Lk^{CHR})

8. Why are the x values in Fig. 3A different in the 4 experiments that are shown? The highest green bars should correspond to the "0 topoisomer" in all 4 experiments and all other topoisomers should have similar integer linking numbers. The axes in these plots should be clearly labeled - both x and y labels are missing here.

We already explained why the plot bars do not have to coincide with integer x-values of ΔLk (point 5). Subsequently, x-values of different minichromosomes are also different because their DNA lengths are not identical. The basis for this expected misalignment is explained in the results section:

....."The above analysis of individual colonies also showed that although different minichromosomes had nearly the same ΔLk , the positions of their Lk topoisomers in the gel did not have the same phasing with respect to Lk^0 (Fig. 3a). This misalignment of the Lk topoisomers occurred because the nucleosomal DNA fragments inserted had different lengths (mean 156 bp \pm SD 8 bp), and Lk phasing occurs only when the length differences are multiples of the helical repeat of DNA ($h \approx 10.5 \text{ bp}$)²⁴ "....

9. The x label of Fig. 1E, "Lk units", is a bit confusing. I suppose this axis actually indicates $\Delta Lk = Lk - Lk_0$, where Lk_0 is the linking number of the relaxed circle ($Lk_0 = L / h$) and Lk is the linking number of the given topoisomer. Please clarify what "Lk units" means, since Lk is a dimensionless quantity, so it doesn't have units of measurement - which makes "Lk units" confusing. Maybe a better label would be "Topoisomer spot (ΔLk)"?

Certainly, "Lk units" was ambiguous. We corrected the x-label by " ΔLk " in all the plots.

10. Please explain in more details how exactly ΔLk was computed. It seems to me that there are different ways to estimate this, e.g. more rigorous by computing its probability distribution as explained at point 2, or more arbitrary by subtracting different quantities, e.g. the means of the 2 distributions, the medians of the 2 distributions, the two most abundant topoisomers, or the fitted Gaussian means - which seems to be the preferred way of the authors. I don't understand why this is the correct way to estimate ΔLk , and consequently I can't appreciate the significance of the obtained result $\Delta Lk/\text{nucleosome} = -1.26$

The new methods subsection and supplementary fig 2 clarifies that ΔLk is calculated by subtracting the means of the distributions.

11. In all the estimations, the authors assume that a fixed number of nucleosomes are found on every plasmid. For example, from Fig. 1 it is obtained that $\Delta Lk = -5.8$, and it is assumed that all plasmids will contain 6 nucleosomes (including 1 centromeric nucleosome). I believe nucleosomes are more dynamic than assumed here, and some plasmids may contain 6 nucleosomes while other only 5 nucleosomes. So how can one eliminate this possibility, which could also generate $\Delta Lk = -5.8$: each nucleosome stabilizes $\Delta Lk/\text{nucleosome} = -1$ but some plasmids contain only 5 nucleosomes, and most of them contain 6 nucleosomes, giving an average of $\Delta Lk/\text{plasmid} = -5.8$?

We constructed YCP1.3 precisely to minimize plausible uncertainties in nucleosome number and dynamics. Our assumption that most or virtually all YCP1.3 minichromosomes have the occupancy described in Fig 1a is supported by several observations. The position of nucleosomes I to V revealed by the MNase digestion is the same to that found in their original genomic loci (TRP1-ARS1). The topology of YCP1.3 ($\Delta Lk -5.8$) is not altered when the minichromosome is solubilized from lysates of unfixed cells, when yeast cells are cultured in rich medium and synthetic drop-out medium; and when YCP1.3 is hosted in yeast cells with reduced topoisomerase activity ($\Delta top1$ and $\Delta top1 top2-ts$). These observations indicate that the chromatin architecture of YCP1.3 is quite stable. However, we do agree with the reviewer in that structural variabilities in a small fraction of YCP1.3 cannot be excluded. But, in this respect, we have explained in the revision (results and discussion sections) that the probability of assembling one more nucleosome is remote. Firstly, there is no sufficient space. Second, reorganization of nucleosome positions is unlikely since the TRP promoter, ARS1, and CEN2 are structural boundaries. On the other hand, we have explained also in the revision (results and discussion sections) that, if nucleosomes were missing, the prospect that $\Delta Lk/\text{nucleosome}$ is about -1.0 would then be even more remote because the absolute $\Delta Lk/\text{nucleosome}$ would result larger than 1.28. Lastly, we have also stressed in the revision that the results obtained after inserting the mononucleosome library further reduces the likelihood that $\Delta Lk/\text{nucleosome} \approx -1$.

Reviewer #2

The authors reexamine a fundamental question in chromatin biology: what is the linking number of DNA containing a nucleosome. Using their published quenching method of capturing plasmids at their in vivo supercoiling state, they determine the linking number for a single nucleosome in their plasmid as -1.28 . They further confirm this number by adding an additional nucleosome containing sequence generated from MNase digest of yeast chromatin. Both with individual clones from the genomic inserts and by pooling all clones, they arrive at similar linking number around -1.26 . They then recalculate the writhe from a nucleosome structure and arrive at -1.53 . So the combination of this writhe value and overtwisting of DNA on nucleosome surface ($+0.2$) leads to a theoretical expectation of -1.3 which is very close to the experimental values obtained by them (-1.26), thus resolving the linking number paradox.

This paper presents well designed experiments answering a long-standing question. The paper is also written well, putting the new results in the proper historical context. I have some minor comments the authors need to address:

We thank the referee for appreciating that our well-designed experiments provide an answer to this long-standing question.

1. The authors need to address in the introduction if the writhe for nucleosome has been determined before, and if not that this also needs to be accounted for to obtain a theoretical linking number change. For example in the introduction, the authors state "the superhelical turning of DNA around a histone octamer ($\Delta Wr \approx -1.7$) predicts that a nucleosome should stabilize about -1.7 units of ΔLk ", but in discussion, they say that writhe has to be calculated. They need to reconcile these two statements in the introduction itself.

Certainly, our statements on historical estimations of Wr were not coherent. We have revised the introduction (paragraphs 2 and 3) and discussion (paragraph 5) to amend this. Namely, following the initial view that DNA Wr in nucleosomes could be close to -2 , early structures of the core particle lead to calculate that Wr is about -1.5 (Le Bret, 1988). This value was reiterated in the reviews of Prunell (1998) and Bates-Maxwell (2005). In supplementary figure 4, we compute the Wr for different degrees of superhelical turning of nucleosomal DNA.

2. In figure 3C, the orange curve seems to be a bimodal distribution, implying a population of nucleosome sequences may have delta $Lk > -1.26$. The authors should fit a bimodal distribution to the orange curve and determine the two mean positions and the

resulting two values of linking number changes for the population.

We realize that the pooled distributions might seem at first sight bimodal. However, this is not the case. The protrusions appear for another reason, as we explained in the revised text:

"....These overlapped signals presented small protrusions (Fig. 3c, green and orange), which could suggest that the pooled Lk distributions were bimodal. This outcome would occur if near one half the nucleosomes of the library constrained $\Delta Lk \approx -1.0$ and the other half $\Delta Lk \approx -1.5$. However, this scenario is not consistent with the data of individual nucleosomes, all which constrained ΔLk values between -1.2 and -1.3 (Fig. 3a). Actually, these protuberances were expected for another reason. They appeared because the different lengths of the DNA inserts were not equally represented (Fig. 2d) and thus so was the phasing of their corresponding LK topoisomers. Accordingly, small protrusions occurred also in the pool of relaxed DNAs. The fact that the pooled Lk distributions presented a dispersion similar to that of individual Lk distributions further substantiated that the pooled samples were essentially monomodal...."

3. In Figure 3A, four representative clones are shown. The authors should indicate which nucleosome position these clones belong to and if they are fuzzy vs. stable. If all four representative clones are similar (for example all are gene body nucleosomes), they should show 2D gels for representative +1, 0, -1 and fuzzy nucleosomes.

As requested, we have indicated in the revised figure 3a the nucleosome attributes of the individual clones. As requested, we included also the analysis of a new colony to have thus representation of +1, 0, -1 and fuzzy vs stable nucleosomes. The main text was revised as follows:

"....Analysis of individual colonies of the library revealed that the minichromosomes had ΔLk values in the range of -7.0 to -7.1 (Fig. 3a). Therefore, relative to the ΔLk of -5.81 stabilized by YCp1.3, the inserted nucleosomes produced $\Delta\Delta Lk$ of -1.2 to -1.3 units. These values were consistent with the average ΔLk of -1.28 per nucleosome calculated for YCp1.3 (Fig. 1e). Moreover, the five clones analyzed in Fig. 3a represented nucleosomes of distinct allocation relative to TSS (-1 , $+1$, $>+1$) and different positional stability (fuzzy or stable). Therefore, these nucleosomes stabilized comparable ΔLk values irrespective of the nucleosome category...."

Reviewer #3

This manuscript addresses the long-known discrepancy between the nucleosome core structure known from X-ray crystal studies and DNA topology measurements in native covalently closed circular minichromosomes known as the "linking number paradox". Initially, studies of SV40 virus minichromosomes containing chains of 22-26 nucleosomes showed that the observed nucleosome-imposed changes in the topological invariant (ΔLk) of about -1 per nucleosome are substantially less than -1.7 predicted from the known nucleosome core structure. Subsequently, studies in biochemically defined in vitro reconstituted mini-circles showed that this difference is mostly due to the linker DNA length and conformation. However, the question of actual ΔLk values generated by native minichromosomes of various linker DNA lengths in vivo remains open. In this work, Segura et al. designed a small yeast minichromosome system in which the number of nucleosome positions can be easily determined and quantified. Importantly, they inserted a library of randomly selected yeast mononucleosomes in their construct that allowed them to quantitate a relative gain of ΔLk per one added nucleosome. In this system, they observed a value of $\Delta Lk = -1.26$ per nucleosome which, in addition to the realistic assumptions for DNA double helical twist ($\Delta Tw = +0.2$) and superhelicity ($\Delta Wr = -1.46$) predicted for slightly unfolded mononucleosomes is closer to the ΔLk predicted from the nucleosome core X-ray crystal structure than the earlier SV40 studies. This is an important and novel experimental observation that contributes to understanding DNA topology and conformation in native chromatin.

Overall, this is a high-quality experimental work, which introduces an innovative minichromosome system to study nucleosome DNA topology and reports interesting and potentially important experimental findings. The manuscript is concise and well written. However, I have a significant concern that important structural parameters describing state of DNA in yeast minichromosomes such as Wr , Tw , and nucleosome occupancy are not experimentally determined but rather subjectively selected from literature to match the ΔLk expected to solve the so-called superhelical paradox. In view of the above concern, the primary conclusion about cracking the linking number paradox remains speculative. To ensure its publication, the title and the main conclusions of the manuscript should be changed to properly reflect the data and/or substantial additional experiments should be conducted to determine the actual structure and nucleosome occupancy of the minichromosomes.

We thank the referee for appreciating the quality of the experimental work, the innovative approach and potential importance of the findings. We understand the concern raised by the reviewer. In contrast to the $\Delta Wr (\approx -1.5)$ and $\Delta Tw (\approx +0.2)$ values of the core DNA, we can only subjectively assume the global Wr , Tw , and nucleosome occupancy of the minichromosome. Unfortunately, an accurate and unambiguous quantification of these structural parameters is quite challenging. However, our experimental results could be hardly explained if nucleosomes were each stabilizing ΔLk of about -1.0 , as previously assumed. As requested, we have revised the text (results and discussion sections) to highlight that, whereas our measured ΔLk /nucleosome provides a solution to the

Lk paradox, this value may not be exact since some contribution of other variables cannot be excluded. We changed the title by replacing the term "cracks" by the less categorical "unfolds".

Specific points:

1. My strongest concern is that the actual number of nucleosomes in the minichromosomes should be precisely determined in order to deduce the number of ΔLk per nucleosome. Current MNase data provide decent measurements of nucleosome positioning but not occupancy and structural variations. Native yeast nucleosomes are notorious for being variously unfolded and invading neighboring nucleosome territories, or completely missing (see e.g. Chereji and Morozov, 2014, PNAS 111, 5236-5241) potentially imposing strong and unknown effects on DNA superhelicity. In classic experiments with isolated SV40 minichromosomes, EM observations were crucial in determining the number of nucleosomes and the beads on a string structure of the minichromosome. Here, to corroborate conclusions of this paper, independent measurements of nucleosome occupancy on individual minichromosome should be conducted if possible by some single minichromosome imaging technique (TEM, AFM, super-resolution fluorescence imaging), otherwise measuring of specific ΔLk value per one nucleosome is likely incorrect.

Determining the exact occupancy and conformation of each nucleosome on individual minichromosomes would be ideal, but these experiments are far from trivial. Moreover, imaging techniques can be intrinsically artefactual and produce data hard to extrapolate to global populations. In our case, this problem is even more complex since minichromosomes are tiny and reside in yeast cells. This is why we designed YCP1.3, with few but well-positioned nucleosomes. We have highlighted in the revision that our assumption that most or virtually all YCP1.3 minichromosomes present the organization depicted in Fig 1a is supported firstly by the MNase digestion, which indicates that the position of nucleosomes I to V is the same as that found in the genomic TRP-ARS loci. Secondly, because the topology of the minichromosome ($\Delta Lk -5.8$) is not altered when it is solubilized from lysates of unfixed cells, nor when yeast cells are cultured in rich medium and synthetic drop-out medium, and nor when YCP1.3 is hosted in yeast cells with reduced topoisomerase activities ($\Delta top1$ and $\Delta top1 top2-ts$). All these observations corroborate that YCP1.3 configuration is quite stable. However, as the reviewer points out, the possibility that structural changes may occur in some minichromosomes cannot be excluded (we commented in the revision the reference indicated by the reviewer). In this respect, we explain also in the revised text that the possibility of assembling/crowding more nucleosomes is remote. First, there is no sufficient space. Second, the TRP promoter, ARS1, and CEN2 are structural boundaries that preclude significant reorganization of nucleosome positions. On the other hand, we also explain that, if nucleosomes were missing in some minichromosomes, the resulting absolute ΔLk /nucleosome would be then larger than 1.28. Consequently, the prospect that nucleosomes stabilize ΔLk of close to -1 would be even more remote.

2. YCP1.3 is a transcriptionally active minichromosome with promoter that may contain transcriptional factors and RNA polymerase II that generate torsional tension. This dynamic torsional tension may affect nucleosome structure making nucleosome chains to accumulate additional negative superhelicity distributed to all nucleosomes (including the added library nucleosomes). The effect of transcription should be addressed by experiments in which either RNApolIII or topoisomerases are inhibited before fixing the nucleosomes or the added nucleosome insulated from the transcription effects.

As requested, we have conducted new experiments in which cells were cultured in rich and drop-out media, and others in which topoisomerases were inhibited before fixing the nucleosomes. We found no significant changes in the topology of the minichromosome (new Fig 1f). This observation, along with the lack of unconstrained supercoils in the unfixed minichromosomes (Fig 1g), corroborates the absence of significant dynamic torsional tension that could deviate the ΔLk of the minichromosome.

3. ΔWr prediction per nucleosome (Fig S3) is based on a isolated mononucleosome and does not take into account the linker DNA length and conformation that are likely to change Wr considerably in a nucleosome array as discussed by the authors in the introduction. Modeling of a circle nucleosome chain rather than a single nucleosome is essential for correct prediction of ΔWr .

Modeling the conformational dynamics of an entire minichromosome in order to compute its overall ΔWr is very complex and probably impracticable without having a high-resolution structure of the entire minichromosome. However, in this respect, we explain in the revised text (discussion) why the folding of linker DNA segments do not significantly deviate the overall Wr :

"...A ΔLk of -1.0 per nucleosome could also stand if the minichromosomes were spatially compacted by adopting a strong negative writhe (i.e. $Wr \approx -1.0$). Such folding would imply that DNA linker lengths and the subsequent rotational orientations between adjacent nucleosomes are alike in all minichromosomes. However, the inserted nucleosome library comprised segments of various lengths and the resulting minichromosomes still constrained very similar ΔLk values...."

4. The authors arbitrary consider the effect of partial nucleosome unwrapping but not that of the increased wrapping of linker DNA that resembles chromatosome and is observed in yeast despite the very low level of linker histone (Cole et al., 2016, NAR 44, 573-581). Precise nucleosome mapping with single nucleotide resolution may help in deducing the linker DNA length and the extent of

DNA unwrapping/overwrapping.

As requested, we revised the discussion to consider the chromosome configuration in the calculation of ΔW_r :

"... Nucleosomal W_r is about -1.53 when the core DNA completes 1.65 left-handed superhelical turns around the histone octamer. This conformation corresponds to that of the crystallized nucleosome structure¹ and also to that of chromosomes⁵³, in which the entry and exit DNA linker segments cross in an angle of about 60 degrees that is fixed by histone H1. This W_r value is likely to reflect thus the upper limit of the absolute DNA writhe of nucleosomes in solution. However, yeast has very low level of linker histone⁵⁴, though nuclease digestions support the existence of proto-chromatosome structures⁵⁵....."

5. Abstract, 9th line should read ($\Delta T_w \approx +0.2$), not ($\Delta T_w \approx +2.0$)

Correction done. Thanks.

Reviewer #1 (Remarks to the Author):

This manuscript has improved significantly after the revision. It is an interesting study that reexamined one of the puzzles of chromatin biology, the linking number paradox, and the conclusion was surprising. Contrary to the previous estimations of ΔLk / nucleosome of -1 (obtained experimentally) and -1.7 (predicted from the crystal structure of the nucleosome), the authors find a linking number value of -1.26 per nucleosome.

Overall this is a valuable study, using an interesting experimental design, which uncovers important experimental findings.

Although the authors have satisfactorily addressed most of my concerns, a few clarifications may still improve the quality of the paper:

1. Regarding my questions about why we see multiple topoisomers in all gels and why their intensities are fitted using a Gaussian distribution, the authors say that "the energy difference between the Lk topoisomers is less than the thermal energy, therefore, Lk distributions tend to be Gaussian". I agree that because the thermal energy is greater than the difference between the energies corresponding to different Lk topoisomers, one expects to see multiple topoisomers in the gels, but that does not imply that the distribution of their intensities is Gaussian. I expect that the energy associated with each topoisomer is quadratic in ΔLk (from a simple Taylor expansion around the energy minimum), and assuming thermal equilibrium, a Boltzmann distribution of the possible topoisomers would generate a Gaussian distribution. Please correct the explanation about why the observed distribution is Gaussian in a more rigorous way.

2. Since the quantification of the intensities in Fig. 3b is crucial for the conclusion of the paper, and the result of $\Delta Lk = -7.06$ seems very precise (2 exact decimals) but very close to the "wrong" prediction (-6.81) of the model that assumes $\Delta Lk/nuc = -1$, I wonder if these small differences are within the experimental error or not. Unfortunately, the description of the procedure of quantifying the intensities from the gels describes very vaguely: "Non-saturated signals of Lk topoisomers were quantified with ImageJ". Also, it is not clear whether the experiment shown in Fig. 3b,c was performed a single time or replicate experiments were used to confirm the result and to test its reproducibility. For me it seems that if one tries to quantify the intensities in Fig. 3b, it is really difficult to decide if the orange distribution from Fig. 3c should be shifted by 7, or 6.8, or another number, relative to the green distribution. To clarify how exactly the authors are able to get this high precision for the x values in Fig. 3c, it would be very useful to see a section in the Supplementary Information explaining how exactly this gel was analyzed. This gel is clearly different from the usual

gels containing distinct topoisomer spots, where the identity of each spot is obvious, so the sentence that "Lk topoisomers were quantified with ImageJ" is not sufficient in this case.

Minor point:

The authors should use a consistent way of separating the decimals from the integer part using a dot instead of a comma (see e.g. red numbers from Fig. 3).

Reviewer #2 (Remarks to the Author):

The authors have addressed all my comments.

Reviewer #3 (Remarks to the Author):

In the revised manuscript, Segura et al. carefully and constructively addressed most criticism and technical concerns by the reviewers. In particular, they experimentally addressed my concern about remaining torsional tension and, though they were not able to exclude the possibility of altered nucleosome number or chromatin structure in their minichromosomes, they provide some arguments against such possibilities.

Overall, the revisions and responses to the reviewers have strengthened my opinion that this is a high-quality experimental work, which introduces an innovative minichromosome system to study nucleosome DNA topology and reports interesting and important experimental findings.

I just suggest a few minor corrections before the final publication:

P. 5, 130: "...at the same temperature (26oC) used to fix the Lk of the YCp1.3 minichromosome in vivo". In fact, the cells were fixed at lower temperature with freezing ethanol. This statement should be corrected e.g. "...at the same temperature (26oC) used to generate the Lk of the YCp1.3 minichromosome in vivo"

P. 6-138 “We subtracted thus these...” – delete “thus”

P. 8-213 and elsewhere in the text: “buLk” – typo (capital L and italics)

P. 11-321 “elucidates the linking number paradox”: this statement reads ambiguously on whether this work clarifies the paradox or exposes the paradox. I suggest saying “provides a plausible explanation for the linking number paradox”.

POINT TO POINT RESPONSE TO REVIEWERS' COMMENTS:

Reviewer #1 (Remarks to the Author):

This manuscript has improved significantly after the revision. It is an interesting study that reexamined one of the puzzles of chromatin biology, the linking number paradox, and the conclusion was surprising. Contrary to the previous estimations of ΔLk / nucleosome of -1 (obtained experimentally) and -1.7 (predicted from the crystal structure of the nucleosome), the authors find a linking number value of -1.26 per nucleosome.

Overall this is a valuable study, using an interesting experimental design, which uncovers important experimental findings.

Although the authors have satisfactorily addressed most of my concerns, a few clarifications may still improve the quality of the paper:

1. Regarding my questions about why we see multiple topoisomers in all gels and why their intensities are fitted using a Gaussian distribution, the authors say that "the energy difference between the Lk topoisomers is less than the thermal energy, therefore, Lk distributions tend to be Gaussian". I agree that because the thermal energy is greater than the difference between the energies corresponding to different Lk topoisomers, one expects to see multiple topoisomers in the gels, but that does not imply that the distribution of their intensities is Gaussian. I expect that the energy associated with each topoisomer is quadratic in ΔLk (from a simple Taylor expansion around the energy minimum), and assuming thermal equilibrium, a Boltzmann distribution of the possible topoisomers would generate a Gaussian distribution. Please correct the explanation about why the observed distribution is Gaussian in a more rigorous way.

RESPONSE: As requested, we used the more adequate term "Boltzmann distribution".

2. Since the quantification of the intensities in Fig. 3b is crucial for the conclusion of the paper, and the result of $\Delta Lk = -7.06$ seems very precise (2 exact decimals) but very close to the "wrong" prediction (-6.81) of the model that assumes $\Delta Lk/nuc = -1$, I wonder if these small differences are within the experimental error or not. Unfortunately, the description of the procedure of quantifying the intensities from the gels describes very vaguely: "Non-saturated signals of Lk topoisomers were quantified with ImageJ". Also, it is not clear whether the experiment shown in Fig. 3b,c was performed a single time or replicate experiments were used to confirm the result and to test its reproducibility. For me it seems that if one tries to quantify the intensities in Fig. 3b, it is really difficult to decide if the orange distribution from Fig. 3c should be shifted by 7, or 6.8, or another number, relative to the green distribution. To clarify how exactly the authors are able to get this high precision for the x values in Fig. 3c, it would be very useful to see a section in the Supplementary Information explaining how exactly this gel was analyzed. This gel is clearly different from the usual gels containing distinct topoisomer spots, where the identity of each spot is obvious, so the sentence that "Lk topoisomers were quantified with ImageJ" is not sufficient in this case.

RESPONSE: As requested, we revised the methods section and added Supplementary Fig. 4 to explain how the gel in figure 3c was analyzed. Namely, pooled Lk distributions were quantified into bins and the ΔLk value of their means was calculated by interpolation with Lk markers. We also indicated in the results and figure 3c that the ΔLk values so obtained were reproduced after four replicate experiments ($\Delta Lk=7.07\pm 0.02$) and consistent with that of the individual clones analyzed in figure 3a.

Minor point:

The authors should use a consistent way of separating the decimals from the integer part using a dot instead of a comma (see e.g. red numbers from Fig. 3).

DONE

Reviewer #2 (Remarks to the Author):

The authors have addressed all my comments.

Reviewer #3 (Remarks to the Author):

In the revised manuscript, Segura et al. carefully and constructively addressed most criticism and technical concerns by the reviewers. In particular, they experimentally addressed my concern about remaining torsional tension and, though they were not able to exclude the possibility of altered nucleosome number or chromatin structure in their minichromosomes, they provide some arguments against such possibilities.

Overall, the revisions and responses to the reviewers have strengthened my opinion that this is a high-quality experimental work, which introduces an innovative minichromosome system to study nucleosome DNA topology and reports interesting and important experimental findings.

I just suggest a few minor corrections before the final publication:

P. 5, 130: "...at the same temperature (26oC) used to fix the Lk of the YCp1.3 minichromosome in vivo". In fact, the cells were fixed at lower temperature with freezing ethanol. This statement should be corrected e.g. "...at the same temperature (26oC) used to generate the Lk of the YCp1.3 minichromosome in vivo"

DONE

P. 6-138 "We subtracted thus these..." – delete "thus"

DONE

P. 8-213 and elsewhere in the text: "buLk" – typo (capital L and italics)

DONE

P. 11-321 "elucidates the linking number paradox": this statement reads ambiguously on whether this work clarifies the paradox or exposes the paradox. I suggest saying "provides a plausible explanation for the linking number paradox".

DONE